# Dynamic Matching Utilizing Latent Factor Modeling

## Abstract

This paper investigates the supply-demand matching problem on dynamic platforms, focusing on optimizing matching strategies by learning workers' attributes when their types are uncertain and constantly changing. To address this problem, we introduce a latent factor model and a multi-centroid grouping penalty mechanism to predict latent factors of workers and perform dynamic matching. Our approach operates in two stages: the first stage fits latent feature vectors for workers and jobs and groups them using historical data; the second stage utilizes these latent features for dynamic matching. Our research demonstrates that the introduced model can adapt to the dynamic changes of the platform with good predictive consistency and group robustness, and improves overall operational benefit through continuous optimization of matching results. We provide simulation experiments and a real case study using kidney exchange data and compare our model with a point process model to show that our approach performs well on dynamic platform matching problems.

**Keywords:** Dynamic matching, Learning, Two-sided market, Bi-clustering, Label aggregation

## 1 Introduction

In the era of digital platforms, the two-sided matching market holds an important position in microeconomics Roth & Sotomayor (1990). The challenge of dynamically matching supply and demand while continuously learning user attributes is both important and complex. Online platforms such as labor markets, e-commerce websites, and fashion retail platforms constantly face the dual challenges of optimizing matching and improving user preferences and ability understanding. This parallel process of matching and learning is crucial for improving user satisfaction and platform efficiency. Moreover, dynamic matching problems also have important applications in the medical field Roth et al. (2004; 2005; 2007) Ashlagi et al. (2013) Ünver (2010), such as how to efficiently match donors and recipients in organ transplantation to improve transplant success rates and patient survival rates, which also requires balancing immediate benefits and long-term learning under limited resources. These practical application scenarios all emphasize the importance of effective matching strategies.

A platform that needs to balance exploration and utilization must find a balance between maximizing matching value (utilization) and continuously learning about new participants to match them (exploration) effectively. Using the terminology of the online labor market, the demand side is referred to as the worker, and the supply side is the job. Under the assumption that supply is limited, matching a supply unit to a user makes it unavailable to others. The platform knows the job type but must learn the unknown worker type through ongoing interactions. Workers leave after completing a certain amount of work. Assuming the platform has system-level knowledge of worker and job arrival rates, as well as the expected benefits from matching specific workers with specific jobs, the goal is to maximize the steady-state cumulative rate of return.

This article presents a model using latent factor grouping penalties to optimize supply-demand matching on dynamic platforms. It addresses the fluctuating nature of job types and worker categories by deeply analyzing their relationships, thereby enhancing platform efficiency. The model operates in two phases: initially fitting latent factors from observed matches, and then using these factors to identify skilled workers for smart matching.

In the first stage, we train these latent factors through historical data, and each job and worker is assigned a latent vector based on their historical performance. These vectors capture the complexity of work and the ability of workers to handle different types of work. For example, the latent vector of a job may highly emphasize technical skills, while the latent vector of a worker indicates their proficiency in that skill. In the second stage, the model identifies which workers have higher work efficiency by analyzing the matching degree between the latent vectors of work and workers. In this process, we introduced a multi-center grouping penalty mechanism, which not only effectively clusters similar jobs and workers, but also controls model complexity through penalty terms to prevent overfitting.

Our research indicates that our model provides a new solution for dynamic platform matching problems, which can adapt to the dynamic changes of the platform and improve the accuracy and efficiency of matching through continuous learning and optimization. The successful application of this method is expected to greatly enhance the platform's operational effectiveness and user satisfaction.

## 2 PROBLEM FORMULATION

In this paper, we study a supply-demand matching problem on a dynamic platform in Johari et al. (2020). We follow the setting in Johari et al. (2020) to represent the supply side with jobs and the demand side with workers. Every job and worker belongs to a specific type and the platform needs to match workers and jobs at each time step ($t = 0, 1, 2, \dots$) based on their types. We assume that the job types on the platform are a finite set $J$ and the worker types are also a finite set $I$. Assume a system with fixed sets of job types $J$ and worker types $I$, utilizing a continuum model with infinitely many workers and jobs to reflect a sufficiently large dynamic market.

Formally, fix a distribution $\rho$ over worker types, i.e. $\rho_i > 0$, $\forall i \in I$ such that $\sum_{i \in I} \rho_i = 1$. We assume that the system initially starts empty before $t = 0$, and in each period $t = 0, ..., N - 1$, a mass $1/N$ of workers arrives in the system. (In what follows we ultimately consider a steady-state analysis of the dynamical system and initial conditions will be irrelevant.) Each worker is of type $i'$ with probability $\rho_{i'}$; each worker's type and arrival time are independent

For simplicity and to maintain a finite mass of workers at all times, we model a process where workers and jobs regenerate every N periods, known as the worker's lifetime, which the platform recognizes. No further arrivals take place after period N. Instead, each worker subsequently regenerates every N period after their arrival: at a regeneration time, the worker type is resampled from the distribution $\rho$; i.e., the new type is i with probability $\rho_i$, and these regenerations are also independent across workers' type and their arriving time.

Our goal is to predict the attributes of workers based on past matched job types and rewards, and maximize expected rewards by adjusting matching, instead of optimizing realized rewards Dai & Jordan (2021) Che & Koh (2016). Similar to decentralized matching markets Roth & Xing (1997) Roth (2008), our platform ensures that only the attributes of workers and jobs and their matching history affect matching. These matches are independent and not influenced by preferences. We define $\mathbf{R} = \{r_{ij}\}_{m \times n}$ as a label matrix, where $r_{ij}$ is the label for the $i$th worker given by the $j$th job ($i = 1, \dots, m; j = 1, \dots, n$). We assume $r_{ij} \in \{0, 1, \dots, C - 1\}$ for multicategory crowdsourcing with $C$ categories. In practice, a worker will not be matched for all jobs, thus only a subset of $\mathbf{R}$ is observed, and we denote the subset as $\Omega = \{(i, j) : r_{ij} \text{ is observed}\}$. We denote $\mathbf{Z} = (Z_1, Z_2, \dots, Z_m)'$ as the true labels for workers, and if $Z_i = c$, we name the $i$th worker as label-$c$ task. The goal of crowdsourcing is to predict $Z_i$'s via aggregating observed $r_{ij}$'s.

Our model is based on task and worker features:

$$\log \left\{ \frac{P(r_{ij} = Z_i)}{1 - P(r_{ij} = Z_i)} \right\} = \mu_i \nu_j \tag{1}$$

where $\mu_i \in (0, \infty)$ is a parameter measuring the difficulty level of the $i$th task and a larger $\mu_i$ indicates less difficulty of the $i$th task, and $\nu_j \in (-\infty, \infty)$ is a parameter measuring the $j$th worker's ability with a higher value associated with greater ability to label tasks. The probability $P(r_{ij} = Z_i)$ converges to 1 if $\mu_i \nu_j \to \infty$, that is, the chance of correct labeling is high with a suitable job for a highly capable worker.

# 3 A MODEL OF MATCHING WHILE LEARNING

In this section, we will introduce a latent factor model from Xu et al. (2023) and propose a two-stage model for multi-category matching. The first stage is to fit the learned worker attributes with the latent factor model, group workers and jobs, and identify suitable jobs. The second stage is to match workers and jobs dynamically with learned factors and groups. The model process can be seen in Figure.1.

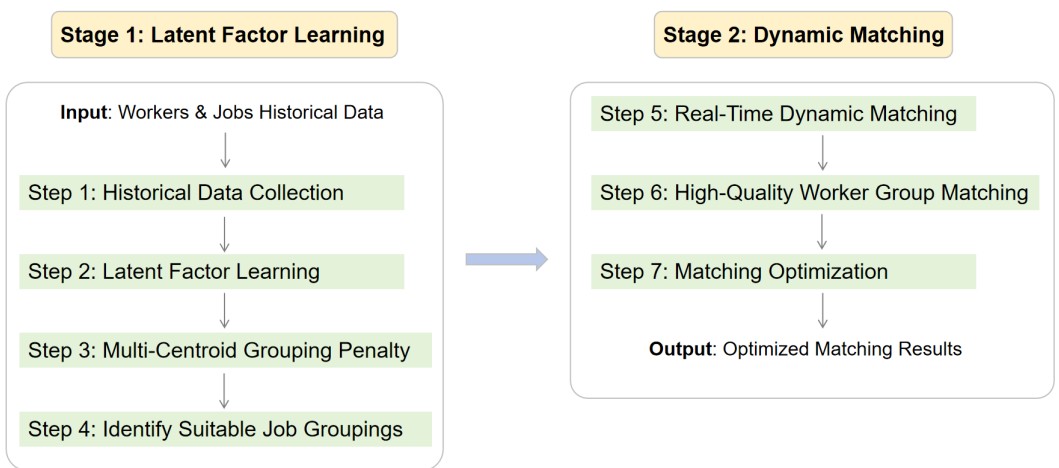

Figure 1: Model Process Overview

**Initialization parameter** Assume that the types of workers entering the platform are limited (there are a total of C types), and that $r_{ij}$ follow a categorical distribution:

$$r_{ij} \sim \text{categorical}(p_{ij}^{(0)}, p_{ij}^{(1)}, \ldots, p_{ij}^{(C-1)}), \quad \sum_{c=0}^{C-1} p_{ij}^{(c)} = 1, \quad (i,j) \in \Omega, \tag{2}$$

where $p_{ij}^{(c)}$ is the probability of $r_{ij}$ equal to $c$. To model multi-category labels more flexibly, we choose to model the probabilities of the observed labels instead of the probability of correctly labeling $\mathbf{1}(r_{ij} = Z_i)$ as in (1).

During the initialization, we set initial $k$-dimensional latent factors $a_i$ and $b_j$ for each worker $i$ and job $j$, reflecting their attributes such as skills and job requirements Whitehill et al. (2009). Normally initialized randomly from $N(0,1)$, these factors can be adjusted based on prior knowledge about workers' skills (if any) to represent their capabilities better, improving matching returns in the initial stage of the market.

Some existing algorithms ignore the heterogeneity of one party in the market Dawid & Skene (2018); Ibrahim et al. (2019); Kim & Ghahramani (2012). To avoid this and fully consider the heterogeneity of workers and jobs, we introduced a full parameterization model:

$$\log \left\{ \frac{p_{ij}^{(c)}}{p_{ij}^{(0)}} \right\} = \mathbf{a}_i' \mathbf{b}_{j,c} - \mathbf{a}_i' \mathbf{b}_{j,0}, \quad c = 1, \ldots, C-1, \quad (i,j) \in \Omega \tag{3}$$

where $\mathbf{a}_i \in \mathbb{R}^k$ is the latent factor for the $i$th worker and $\mathbf{b}_{j,c} \in \mathbb{R}^k$ represents the $j$th job's feedback for the label-$c$ worker (preliminary determination of the value of each dimension corresponding to worker attributes based on the benefits obtained from matching workers and jobs). The inner product $\mathbf{a}_i' \mathbf{b}_{j,c}$ measures the reward of matching the $i$th worker and the label-$c$ job based on the $j$th job's feedback. Contrary to (1), the true labels $Z_i$'s are implicitly embedded in $\mathbf{a}_i$'s, and we assume that workers in the same category are likely to share similar latent features. Therefore, the latent factors of tasks are expected to be clustered, where we use $\mathbf{U} = \{U_i\}_{i=1}^m$ ($U_i \in \{0, 1, \ldots, C-1\}$) to denote the worker memberships. Here task membership $\mathbf{U}$ is a permutation of true labels $\mathbf{Z}$, and we need to recover the correspondence between $\mathbf{U}$ and $\mathbf{Z}$ in the later step.

**Stage One: Potential Factor Learning** Here we introduce an approach to leverage the subgroup structures within workers and jobs. Suppose there are $D$ job groups in the market and the group membership of jobs is denoted by $\mathbf{V} = \{V_j\}_{j=1}^n$ where $V_j \in \{1, 2, \ldots, D\}$. Jobs from the same subgroup tend to have similar feedback for workers, and within-group correlations among jobs' feedback should be taken into consideration. For job subgroup $d \in \{1, 2, \ldots, D\}$, we introduce a set of rotation matrices

$$\mathbf{O}^{(d)} = \{\mathbf{O}_c^{(d)} \mid \mathbf{O}_c^{(d)'}\mathbf{O}_c^{(d)} = \mathbf{I}_{k \times k}, \mathbf{O}_c^{(d)} \in \mathbb{R}^{k \times k}, c = 0, 1, \ldots, C - 1\}, \tag{4}$$

to align the latent factors of the $d$th job subgroup with the latent factors of each worker category. Specifically, for the $j$th job belonging to the $d$th subgroup, its feedback for the label-$c$ workers is modeled based on rotating its feedback for the reference category (label-0 workers are treated as a reference in default in this paper) via the rotation matrix $\mathbf{O}_c^{(d)}$:

$$\mathbf{b}_{j,c} = \mathbf{O}_c^{(d)}\mathbf{b}_j, \quad c = 0, 1, \ldots, C - 1, \quad d = V_j, \tag{5}$$

where we specify $\mathbf{O}_0^{(d)} = \mathbf{I}_{k \times k}$ for an identifiability purpose such that $\mathbf{b}_j = \mathbf{b}_{j,0}$ represents the $j$th job's feedback for the reference group. We denote all the rotation matrices as $\mathbf{O} = \{\mathbf{O}^{(1)}, \mathbf{O}^{(2)}, \ldots, \mathbf{O}^{(D)}\}$. Based on the modeling in (3), the multicategory crowd labels follow a logistic model:

$$\theta_{ijc} = \log\left\{\frac{P(r_{ij} = c)}{P(r_{ij} = 0)}\right\} = \mathbf{a}_i'(\mathbf{O}_c^{(d)}\mathbf{b}_j - \mathbf{a}_i'\mathbf{b}_j), \quad c = 1, \ldots, C - 1, \quad d = V_j, \quad (i, j) \in \Omega \tag{6}$$

Note that the fitting model (6) does not predict the true label $\mathbf{Z}$ directly, but rather restores the subgroup membership of worker $\mathbf{U}$. Therefore, we aim to restore the correct correspondence between the worker's membership degree $\mathbf{U}$ and the true label $\mathbf{Z}$. To achieve this goal, we identify appropriate job subgroups for each worker category and restore the corresponding relationships based on the labels of suitable jobs. We assign unique labels to each job subgroup and rely on the following assumptions for suitable job groups:

**Assumption 1**.*For each worker category, a subset of jobs is the suitable job set for that worker category if the members of the set have a greater match with workers in that category than other sets. Matching suitable jobs with relevant workers is expected to yield the highest benefits.*

Then, we need to identify high-quality workers. Assume that there are $D$ worker subgroups. Denote $\alpha_I, I = 0, 1, \ldots, C - 1$ as latent factor centroids of jobs, and $\beta_{I,J}, I = 0, 1, \ldots, C - 1, J = 1, 2, \ldots, D$ for worker subgroup $J$ for each job category $I$. In general, we can identify the high-quality worker subgroup and the best profit for the $i^{th}$ job by the following criterion based on Assumption 1:

$$(\hat{c}_i, \hat{d}_i) = \underset{c \in \{0, 1, \ldots, C-1\}, \ d \in \{1, 2, \ldots, D\}}{\arg \max} \alpha_{U_i}'\beta_{c,d}$$

where $\hat{d}_i$ is the high-quality worker group for the $i^{th}$ job and $\hat{c}_i$ is the profit of the $i^{th}$ job by matching with the high-quality worker group $\hat{d}_i$. The maximization can be done by profiling. For each fixed worker group d = 1, 2, ..., $D$, the maximization over c corresponds to aligning the $d^{th}$ group workers' profit with $\alpha_{U_i}$. The maximization over d corresponds to identifying the high-quality worker group for the $i^{th}$ job.

**Stage Two: Dynamic Matching** We apply the above learning process to the entire matching process and consider more general cases. When $t = n, \forall n \geq 0$, assume that the number of newly arrived workers is $n_1$ and the number of newly arrived jobs is $n_2$. The job types belong to 0, 1, ..., $C$ (there are $C$ job types in total) and the worker types belong to 0, 1, ..., $D$ (there are $D$ job types in total). The platform needs to match each worker with an available job.

We first consider the workers who have arrived and are still on the platform at this time. The assumption is that the platform has a history of each worker's work (including the types of jobs they have matched with and their job earnings), so it is necessary to learn the workers' attributes based on these historical records. The historical record of a worker is denoted as H=$((j_1, r_1), \ldots, (j_m, r_m))$, where $j_n$ represents the type of job that this worker has matched at the $nth$ time step in the system since the last regeneration; $r_n$ belongs to $R$ and represents the performance of this worker in the $j_n$ type of job, which, for convenience, can be expressed in terms of revenue.

Through the benefit matrix, the label records of each worker can be obtained from the work history records. Specifically, if the benefit matrix is known, and a worker generates a benefit of 1 when completing job $j$, then label the worker $i$ ($A_{ij} = max_{k \in I} A_{kj}$); In the event that the generated benefit is equal to zero, the worker in question should be assigned the corresponding type label based on the proportion of failure probability when other jobs are transferred to other types of workers. We believe that this allocation method is reasonable and efficient when the sample size is sufficiently large.

Next, it is necessary to consider all workers within the system, including the set of workers $A$ (assuming a quality of $1/N$) that are currently being regenerated and the workers $M$ who have previously arrived in the system. By analyzing the historical records of all workers in $M$, suitable jobs can be identified for each worker type. A threshold is established for the set in $M$. If the mean revenue generated by matching corresponding suitable jobs with distinct worker types reaches this threshold, then these jobs are directly assigned to the corresponding worker type. Let $K$ be the set of all eligible workers. After matching the workers in $K$, the remaining jobs are randomly assigned to workers in $N$ and $M \setminus K$.

Specifically, given the known distribution of workers, denoted by the function $\rho$, the number and type of jobs, and the number of workers equal to the number of available jobs (assuming a sufficiently large number and a strict distribution of $\rho$), we consider random matches between these workers and jobs. Assuming the platform can generate revenue based on the type of workers and jobs, the expected total revenue $W_0$ can be calculated. Furthermore, the expected average benefit per worker $W$ can also be obtained and set as the aforementioned threshold. Additionally, the weights of different job types can be adjusted. For example, the weight of high-difficulty job types can be increased to minimize the likelihood of matching them with low-ability workers.

## 4 Algorithm

### 4.1 Multi-centroid Grouping Penalty

To group workers and jobs, we introduce the multi-centroid grouping penalty from Xu et al. (2023). Denote A = $(a_1, a_2, ..., a_m)$ as collections of latent factors for jobs and B = $(b_1, b_2, ..., b_n)$ as collections of latent factors for workers where $b_i = b_{i,0}$ (default with type-0 jobs as the reference). Suppose that the group membership of workers and jobs is represented by the sets V and U, respectively. In this context, we define the multi-centroid grouping penalty as follows:

$$\mathcal{G}_\lambda(\mathbf{A}, \mathbf{B}, \mathbf{U}, \mathbf{V}) = \lambda \left\{ \|\mathbf{A} - \mathbf{PA}\|_F^2 + \|\mathbf{B} - \mathbf{QB}\|_F^2 \right\}$$

where $\lambda$ is a tuning parameter for penalization, matrices $P$ and $Q$ are projection matrices associated with group memberships U and V by a one-to-one mapping Hoaglin & Welsch (1978) to calculate the subgroup centroids of jobs and workers separately. Specifically, the projection matrices $P$ and $Q$ project each latent factor $a_i$ and $b_j$ onto its corresponding centroids, i.e. $Pa_i = \alpha_{U_i}$ and $Qb_j = \beta_{0,V_j}$. Therefore, the multi-centroid grouping penalty is equivalent to

$$\mathcal{G}_\lambda(\mathbf{A}, \mathbf{B}, \mathbf{U}, \mathbf{V}) = \lambda \left\{ \sum_{I=0}^{C-1} \sum_{\{i:U_i=I\}} \|\mathbf{a}_i - \boldsymbol{\alpha}_I\|^2 + \sum_{J=1}^{D} \sum_{\{j:V_j=J\}} \|\mathbf{b}_j - \boldsymbol{\beta}_{0,J}\|^2 \right\}.$$

By incorporating the multi-centroid grouping penalty, we can estimate latent factors A, B, and group memberships U, and V jointly by minimizing the negative penalized log-likelihood:

$$(\hat{\mathbf{A}}, \hat{\mathbf{B}}, \hat{\mathbf{O}}, \hat{\mathbf{U}}, \hat{\mathbf{V}}) = \arg \min_{\mathbf{A}, \mathbf{B}, \mathbf{O}, \mathbf{U}, \mathbf{V}} \mathcal{L}(\mathbf{A}, \mathbf{B}, \mathbf{O}) + \mathcal{G}_\lambda(\mathbf{A}, \mathbf{B}, \mathbf{U}, \mathbf{V}). \tag{7}$$

We introduce an alternating minimization algorithm to minimize this joint loss function. Specifically, at the $(t+1)$th iteration, we update $(\hat{\mathbf{A}}, \hat{\mathbf{B}}, \hat{\mathbf{O}})$ sequentially given the estimation $(\hat{\mathbf{U}}^{(t)}, \hat{\mathbf{V}}^{(t)})$ as follows:

$$\hat{\mathbf{A}}^{(t+1)} = \arg\min_{\mathbf{A}} \mathcal{L}(\mathbf{A}, \hat{\mathbf{B}}^{(t)}, \hat{\mathbf{O}}^{(t)}) + \lambda \|\mathbf{A} - \hat{\mathbf{U}}^{(t)}\mathbf{A}\|_F^2,$$

$$\hat{\mathbf{B}}^{(t+1)} = \arg\min_{\mathbf{B}} \mathcal{L}(\hat{\mathbf{A}}^{(t+1)}, \mathbf{B}, \hat{\mathbf{O}}^{(t)}) + \lambda \|\mathbf{B} - \hat{\mathbf{V}}^{(t)}\mathbf{B}\|_F^2,$$

$$\hat{\mathbf{O}}^{(t+1)} = \arg\min_{\mathbf{O}} \mathcal{L}(\hat{\mathbf{A}}^{(t+1)}, \hat{\mathbf{B}}^{(t+1)}, \mathbf{O}),$$

where the minimization regarding $\mathbf{O}$ is realized via performing Cayley transformation Wen & Yin (2013) which guarantees the orthogonal constraints are held in the iterations. Specifically, we introduce $\mathbf{G} = \{\mathbf{G}_0^{(1)}, \mathbf{G}_1^{(1)}, \ldots, \mathbf{G}_{C-1}^{(1)}, \ldots, \mathbf{G}_{C-1}^{(D)}\}$ as gradients of $\mathcal{L}$ with respect to $\mathbf{O}$, where

$$\mathbf{G}_c^{(d)} = \frac{\partial \mathcal{L}}{\partial \mathbf{O}_c^{(d)}}.$$

We then define a set of matrices $\mathbf{S} = \{\mathbf{S}_0^{(1)}, \mathbf{S}_1^{(1)}, \ldots, \mathbf{S}_{C-1}^{(1)}, \ldots, \mathbf{S}_{C-1}^{(D)}\}$, where

$$\mathbf{S}_c^{(d)} = \mathbf{G}_c^{(d)}\mathbf{O}_c^{(d')\prime} - \mathbf{O}_c^{(d)}\mathbf{G}_c^{(d')\prime}.$$

Then $\mathbf{O}_c^{(d)}$'s are updated following the Cayley transformation:

$$\mathbf{O}_c^{(d)} \leftarrow (\mathbf{I} + \frac{\eta}{2}\mathbf{S}_c^{(d)})^{-1}(\mathbf{I} - \frac{\eta}{2}\mathbf{S}_c^{(d)})\mathbf{O}_c^{(d)}, \quad d = 1, \ldots, D, \quad c = 1, \ldots, C-1,$$

where $\eta$ is a positive learning rate. We update $\mathbf{O}_c^{(d)}$'s iteratively until the algorithm converges. It is noticeable that $\mathbf{O}_c^{(d)}$'s remain orthogonal at each iteration for any positive value of $\eta$, and we set $\eta = 0.1$ empirically which leads to a fast convergence while maintaining a low training error. In addition, the update of latent factors $\mathbf{a}_i$'s and $\mathbf{b}_j$'s can be paralleled to speed up the computation. Furthermore, the updates of projection matrices $\mathbf{U}$ and $\mathbf{V}$ are equivalent to cluster task and worker latent factors into $C$ and $D$ subgroups separately. Therefore, we update $\mathbf{U}$ and $\mathbf{V}$ with the K-Means algorithm.

## 5 THEORY

This section develops theoretical guarantees of our proposed mode. These theorems not only demonstrate the soundness of our approach but also reassure users of its applicability in dynamic and complex matching environments. See the Appendix for detailed proof.

**Theorem 1** (Predictive Consistency). *Let $M$ denote the observed label matrix with entries $M_{ij}$ representing the label assigned by worker $j$ to task $i$, and $\Theta$ denote the latent feature matrix. Consider a loss function $L(M, \Theta)$ and a group structure regularization $G(\Theta)$. Define the model optimization problem as:*

$$\hat{\Theta} = \arg\min_{\Theta} \{L(M, \Theta) + \lambda G(\Theta)\},$$

*where $\lambda$ is the regularization parameter. Under suitable regularity conditions on $L$ and $G$, and given appropriate choice of $\lambda$ and sufficiently large sample size $n$, it holds that*

$$\mathbb{P}(\hat{\Theta} \text{ consistently estimates } \Theta^*) \to 1 \text{ as } n \to \infty.$$

**Remark 1.** *The core significance of Theorem 1 is that it ensures that as the sample size of data increases, the model can gradually converge to the true features of users and tasks, ensuring the accuracy of the matching algorithm in dynamic platforms. In the first stage of latent feature learning, historical data is used to fit the latent feature vectors of users and tasks, minimizing estimation errors. By increasing the sample size, the model can gradually improve the accuracy of prediction, which supports the algorithm's ability to continuously improve matching performance in dynamic environments.*

**Theorem 2** (Group Robustness). *Define the true group labels for workers and tasks as $\mathbf{V}$ and $\mathbf{U}$, respectively, with estimated group labels $\hat{\mathbf{V}}$ and $\hat{\mathbf{U}}$. The group robustness measure $R(\mathbf{V}, \mathbf{U})$ is defined as:*

$$R(\mathbf{V}, \mathbf{U}) = \frac{1}{n}\sum_{i=1}^{n} \mathbf{1}(\hat{v}_i = u_{\sigma(i)}),$$

*where $\sigma$ is the permutation that maximizes $R$. Under the assumption of adequate model specification and large data limit, it holds that*

$$R(\mathbf{V}, \mathbf{U}) \to 1 \text{ as } n \to \infty.$$

**Remark 2.** *Theorem 2 demonstrates the classification robustness of the model in multi-class matching tasks, and as the sample size increases, the classification error will gradually approach zero. This theorem is directly related to the multi-class task matching in the second stage of the model. The model clusters tasks and users through latent factor models and multi-center grouping penalty mechanisms to ensure the stability and robustness of classification. With the increase of data size, the improvement of group classification accuracy can significantly enhance the matching efficiency in dynamic platforms.*

**Theorem 3** (Convergence Rate). *Define the estimation error $\epsilon(\Theta) = \|\hat{\Theta} - \Theta^*\|$. Under appropriate regularization and sufficient sample size, the convergence rate of the estimation error is:*

$$\epsilon(\Theta) = O\left(\frac{1}{\sqrt{n}}\right),$$

*indicating that the error decreases at the rate proportional to $n^{-1/2}$ as the sample size $n$ increases.*

**Remark 3.** *Theorem 3 quantifies the convergence rate of the model, ensuring that as the sample size increases, the estimation error decreases at the rate of $O(1/\sqrt{n})$, providing a clear expectation for matching performance. This theorem is directly related to the multi-centroid grouping penalty mechanism in the model, which helps to reduce overfitting in the latent feature space, thereby improving matching accuracy. By understanding the convergence rate, platforms can plan better strategies for data accumulation and matching accuracy improvements.*

The above theorems form a robust mathematical framework. They ensure that the model not only captures the complex dynamics of the marketplace but also adapts efficiently to changes, maintaining high levels of accuracy and operational effectiveness.

## 6 NUMERICAL STUDIES

### 6.1 MATCHING SYSTEM SIMULATION WITH QUEUE-BASED SHADOW PRICES

In this section, we simulated an allocation process with queue-based shadow prices. (More comprehensive surveys of dynamic pricing and learning can be seen in den Boer & Zwart (2015) Sauré & Zeevi (2013) Babaioff et al. (2013).) We consider a marketplace with $|\mathcal{I}| = 4$ worker types and $|\mathcal{J}| = 3$ job types. The arrival rates of all the worker types are identical, i.e., $\rho_i = 0.25$ for all $i \in \mathcal{I}$, while the arrival rates of the job types are randomly chosen. The assumptions about the simulated marketplace are described as follows.

**Arrival process.** Time is discrete, $t = 1, 2, \ldots,,$ where we assume that $N = 10, 20, 30, 40$. At the beginning of each time period $t$, a fixed $M_N(i)$ number of workers of type $i$ arrive and they stay for $N$ periods. We choose $M_N(i) = 60/N$ for the different values of $N$ for each $i$, so that, irrespective of $N$, the total number of workers of any type $i$ present in the market at any time $t$ is $M_N(i) \times N = 60$ (making a total of $60 \times |\mathcal{I}| = 60 \times 4 = 240$ workers present in the market at any time).

**Matching process.** In this process, we set up the following benefit matrix, where the last column corresponds to an empty job type.

$$A = \begin{bmatrix} 0.9 & 0.5 & 0.8 & 0 \\ 0.6 & 0.9 & 0.4 & 0 \\ 0.3 & 0.8 & 0.7 & 0 \\ 0.7 & 0.6 & 0.9 & 0 \end{bmatrix}$$

To better utilize our model, we tried to match each worker with the same type of work at every time point in the first half of each process, to obtain high-quality work groups corresponding to different types of workers. In each match, we use the benefit matrix and job type to label workers and record

all labels. Then, the latent factor model is used to infer the predicted labels for each worker, and a more accurate predicted label is selected for subsequent work assignments.

**Simulation output.** We set the performance ratio as the main output, which presents The total rate of reward for each time during the whole process. The result is shown in Figure 2 - 5.

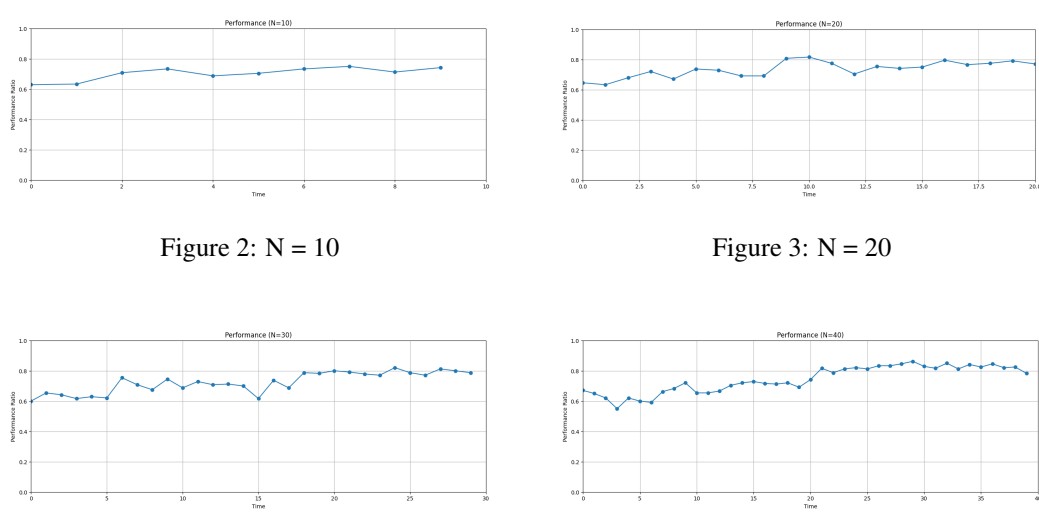

Figure 2: N = 10

Figure 3: N = 20

Figure 4: N = 30

Figure 5: N = 40

It can be seen that the total reward gradually increases over time, but the increase is not significant, which is actually related to the number of worker types and job types.

## 6.2 MATCHING SYSTEM SIMULATION WITH REAL-WORLD DATA ON KIDNEY EXCHANGES

In this section, we use our model to analyze data on kidney exchange in the real world Dai & He (2023) and attempt to improve the success rate of kidney exchange, using data from the Organ Procurement and Transplantation Network (OPTN) as of July 2022 provided by the United Network for Organ Sharing (UNOS). We first collected and organized relevant information on kidney transplantation, including recipient age, recipient age group, recipient gender, recipient blood type, donor age, donor gender, donor blood type, PRA, AMIS, BMIS, DRMIS, HLAMIS (MIS represents the corresponding antigen or HLA mismatch level), and GSTATUS-KI for each kidney transplantation surgery. Our goal is to use these covariates to train a model and predict the compatibility of donors and recipients in kidney transplantation surgery.

We treat each kidney transplant surgery as a task to be predicted, including the corresponding recipient age, recipient age group, recipient gender, recipient blood type, donor age, donor gender, and donor blood type. In other words, we treat the donor-recipient matching group that contains this information as a task—simultaneously using GSTATUS-KI to define the success of transplantation, where 1 represents failure and 0 represents success, thus generating a true label set for judging accuracy.

On the other hand, we use five covariates, PRA, AMIS, BMIS, DRMIS, and HLAMIS, as the workers who label tasks. We calculated the 1/3 and 2/3 percentiles for each covariate, which serve as thresholds for subsequent classification. To accurately reflect the impact of different mismatch indicators on the success rate of transplantation, we designed two classification functions. Specifically, for the PRA indicator, due to its opposite impact direction compared to other indicators, we adopted a special classification logic: when the PRA value is higher than its 2/3th percentile, we consider the result to be more ideal and therefore assign a lower classification label (0). On the contrary, for other covariates, when the covariate value exceeds its 1/3 percentile, we classify it as 1, indicating that the result may not be ideal. This approach ensures that each task has five workers predicting its label and generating a corresponding set of labels. We applied the processed dataset to the model and obtained an accuracy of 60% for matching prediction.

## 6.3 COMPARISON WITH POINT PROCESS MODEL

As a comparison, we introduced the point process model from Perry & Wolfe (2013). Point process models analyze random event occurrences in time or space by defining an intensity function that describes event probability based on time, location, and covariates. These models, used for time series or spatial data, estimate parameters by maximizing likelihood functions to predict event patterns. We try to use this model to analyze kidney transplant data and solve the matching problem between donors and recipients. The idea is as follows.

Firstly, determine the objective of the problem: to predict the compatibility between donors and recipients based on their detailed information. We collected data related to kidney transplantation, including

**Donor-related covariables:** donor blood type, donor age, donor gender, donor BMI, donor hypertension history, donor diabetes history, donor hepatitis C antibody results, donor hepatitis B surface antigen results

**Recipient-related covariables:** recipient blood type, recipient age, recipient gender, recipient BMI, recipient CMV status, recipient HIV serum status, whether the recipient receives dialysis treatment

**Matching related covariables:** donor-recipient ABO matching level, HLA mismatch level, kidney cold ischemia time (hours), total days on the waiting list, transplantation date

**Result covariables:** graft survival Period (the number of days from transplantation to failure/death/final follow-up), whether the graft failed or not

To apply the point process model, we consider the success or failure of kidney transplantation as an event and each transplantation attempt as a directed interaction (donor to recipient). Our goal is to build a predictive model to determine whether successful transplantation will occur within a given time frame.

Specifically, we use a form similar to the Cox proportional strength model to model the interaction (i.e. successful transplantation) as a point process. Assuming the donor is $i$, the recipient is $j$, and the transplantation time is $t$, define the intensity function $\lambda_t(i, j)$

$$\lambda_t(i, j) = \lambda_0(i) \exp(\beta^T x_t(i, j))$$

where $\lambda_0(i)$ is the baseline hazard function for the $i$-th individual. $x_t(i, j)$ is the vector of covariates affecting the $i$-th individual and the $j$-th event at time $t$. $\beta$ is the vector of coefficients.

Next, in the data preprocessing stage, we standardize the date format and encode categorical variables such as gender, history of hypertension, diabetes, and HIV status. Categorical features like blood type are processed using one-hot encoding, and missing values in the data are filled to ensure data integrity.

During model parameter initialization, the initial estimates of model coefficients are set to zero, and the initial weights for the risk model are set to 1. A critical step involves calculating the gradients and Hessians, based on the risk set expression

$$W(\beta, i, j) = W_0(\beta, i) \exp(\Delta X_t(i, j)^T \beta)$$

where $W_0(\beta, i, j)$ represents the baseline risk, and $\Delta X_t(i, j)$ is the dynamic part of the covariates.

In defining the objective function, we optimize the objective function defined by the gradients and Hessians using Newton's method. The specific calculations of gradients and Hessians rely on the formulas

$$\nabla \log PL(\beta) = \sum_{i \in I} (X_i^T \pi_t - E_i)$$

$$\nabla^2 \log PL(\beta, i) = X_i^T (\Pi_{\beta,i} - \pi_{\beta,t,i}^T \pi_{\beta,t,i}) X_i$$

where $\pi_{\beta,t,i} = W(\beta,i,j)/W(\beta,i)$ represents the normalized risk.

Finally, model optimization and evaluation are conducted by minimizing the objective function to find optimal parameters, and the predictive performance of the model is assessed by calculating the ROC curve and AUC score. The specific results we obtained are shown in Figure 6.

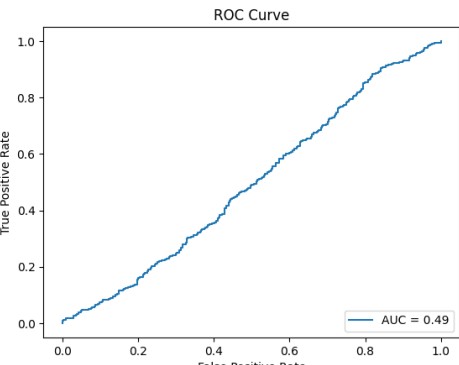

Figure 6: N = 10

By comparison, point process modeling and the matching while learning model have some similarities when addressing kidney transplant matching issues, such as needing to preprocess raw data by encoding and normalization, and estimating parameters by maximizing some form of likelihood function. However, they have significant differences in model structure, dynamic processing, and network effects. Point process modeling, using the Cox proportional intensity model, focuses on time dynamics and event probability, ideal for time-dependent data. It processes dynamic covariates through partial likelihood inference, evaluating complex network interactions like donor-recipient homophily. In contrast, the matching while learning model, employing a latent factor model with multi-centroid grouping, suits scenarios requiring dynamic updates in worker-job groupings. It updates attributes and matches quality based on historical data, handling complex network effects less extensively.

## 7 CONCLUSION

This article uses the latent factor model to solve the dynamic supply problem in market matching, focusing on how to reasonably use the two stages of the model for this problem, as well as considering the impact of time in the matching process. Our application enables the model to adapt to the dynamic changes of the platform and continuously optimize the matching efficiency and matching results. Experiments show that this method is suitable for the matching problem of multi-class workers and jobs, and the results are better than other methods. The advantage of this method is that it can handle markets with more individual categories, the number of which is theoretically unlimited. However, in practical situations, as the types of workers and jobs in the market increase, the accuracy of this method needs to be improved, which can serve as a future research direction.

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

## A  APPENDIX

### A.1  PROOF OF THEOREM 1 AND THEOREM 3

Define the loss function $L(M, \Theta)$ as the likelihood function for the observed data $M$ and the parameter $\Theta$. Assume $L$ is convex and continuously differentiable for all $\Theta$. The regularization term $G(\Theta)$ is also convex and continuously differentiable for all $\Theta$, used to introduce prior knowledge of group structures.

Firstly, we assume that the estimator $\hat{\Theta}$ is obtained from the following optimization problem as sample size $n$ increases:

$$\hat{\Theta} = \arg\min_{\Theta}\{L(M, \Theta) + \lambda G(\Theta)\}$$

Since $L$ is convex, we can use Taylor expansion to approximate the local behavior of $L$ at $\Theta^*$:

$$L(M, \Theta) \approx L(M, \Theta^*) + \nabla L(M, \Theta^*)^T(\Theta - \Theta^*) + \frac{1}{2}(\Theta - \Theta^*)^T\nabla^2 L(M, \Theta^*)(\Theta - \Theta^*).$$

At $\Theta^*$, $\nabla L(M, \Theta^*) = 0$ as $\Theta^*$ is the minimum point of $L$.

According to the Law of Large Numbers, we know that the average of the estimators converges to the expected value, i.e.,

$$\bar{\Theta} \xrightarrow{p} E[\Theta].$$

Using the Central Limit Theorem, the distribution of the estimators approaches a normal distribution:

$$\sqrt{n}(\bar{\Theta} - E[\Theta]) \xrightarrow{d} \mathcal{N}(0, \Sigma),$$

where $\Sigma$ can be calculated from the inverse of the Fisher information matrix $\mathcal{I}(\Theta^*)$ at $\Theta^*$:

$$\Sigma = \mathcal{I}(\Theta^*)^{-1}, \quad \mathcal{I}(\Theta) = E[-\nabla^2 L(M, \Theta)].$$

Appropriately choose $\lambda_n$ to ensure $\lambda_n \to 0$ and $n\lambda_n \to \infty$, so that the regularization term is sufficiently small not to affect consistency but enough to maintain model complexity. Specifically, we can take $\lambda_n = \frac{1}{\sqrt{n}}$.

With the choice of $\lambda_n$ as mentioned, we can show:

$$\|\hat{\Theta} - \Theta^*\| = O_p(n^{-1/2}),$$

meaning that the decrease in estimation error rate is proportional to $n^{-1/2}$, which is the standard rate of convergence for parameter estimation.

### A.2 PROOF OF THEOREM 2

In considering the proof of group robustness, the first step is to define an appropriate clustering algorithm. In this proof, we choose the spectral clustering algorithm because it effectively handles data groups formed in latent feature spaces that are non-spherical or linearly inseparable.

We choose to use the k-means clustering algorithm because it performs well in handling data with well-defined group centers. The algorithm iterates the following steps until convergence:

1. Randomly select $C$ initial centers.
2. Assign each point to the nearest center, forming $C$ groups.
3. Update each group's center to the mean of all points in that group.

In spectral clustering, it's crucial to ensure that different groups are separable in the latent space. We use concentration inequalities, such as Hoeffding's inequality, to ensure that the distances between different groups are statistically significant.

For independent and identically distributed random variables $X_1, X_2, ..., X_n$ with expectation $\mu$, we have:

$$P(|\bar{X} - \mu| \geq t) \leq 2\exp(-2nt^2/\text{range}(X)^2),$$

where $\bar{X}$ is the sample mean, and $t$ is any positive number. In our scenario, $X_i$ might be a measure of some latent feature dimension, and this inequality helps bind the probability of intra-group consistency and inter-group separation.

After clustering, we compare the estimated group labels $\hat{U}$ and $\hat{V}$ to the true group labels $U$ and $V$. The classification error bound is defined by calculating the rate of inconsistencies between the estimated and true labels.

Let $R(\hat{V}, \hat{U})$ be the proportion of correctly classified instances, then:

$$R(\hat{V}, \hat{U}) = \frac{1}{n}\sum_{i=1}^{n} 1(\hat{v}_i = u_{\sigma(i)}),$$

where $\sigma$ is a permutation that maximizes the matching accuracy between estimated and true labels. And as $n$ approaches positive infinity, $R(\hat{V}, \hat{U})$ goes to 1.

