# OpenReview forum: "Dynamic Matching Utilizing Latent Factor Modeling"
_ICLR.cc/2025/Conference — ICLR 2025 Conference Withdrawn Submission_

### Official Review · Reviewer_wXHg · 2024-10-24

**Soundness:** 2
**Presentation:** 3
**Contribution:** 2
**Rating:** 5
**Confidence:** 3

**Summary:**

To address the issue of dynamic supply in market matching, this manuscript proposes a two-phase adaptive matching strategy that responds to changes within the platform. This strategy introduces a latent factor model and a multi-centroid grouping penalty mechanism to predict latent factors of workers. Furthermore, the paper conducts a simulation experiment involving kidney exchange and performs a comparative analysis with the point process model, demonstrating the proposed method's robust and predictive capabilities.

**Strengths:**

1.The manuscript exhibits a clear structure, adheres to standard conventions in the use of formulas, and presents a rigorous and lucid argumentative process, complemented by a rich content base.
2.The dynamic study of the supply-demand relationship, exemplified through the worker-job framework, possesses significant practical applicability.
3.A two-stage dynamic matching method utilizing latent factor modeling is proposed, featuring a clear and ingeniously designed structure. The system is applied to kidney exchange matching, effectively demonstrating the practical value of this research.

**Weaknesses:**

1.The manuscript lacks a cohesive overview of related work.
2.There are inconsistencies in the use of nouns and variables throughout the manuscript. For instance, in Section 4.1, "MULTI-CENTROID GROUPING PENALTY," A denotes collections of latent factors for jobs, whereas in Section 3, "A MODEL OF MATCHING WHILE LEARNING," ai refers to the latent factor for the i-th worker. Additionally, Section 3 alternates between the terms "benefit matrix" and "reward matrix." These discrepancies diminish the clarity of the manuscript and may confuse readers.
3.The manuscript contains several minor spelling and indentation errors. Further review and revision would enhance the overall readability of the paper.

**Questions:**

1.The authors assume that both workers and jobs regenerate after every N periods. However, in reality, both may disappear after random intervals; for instance, new job positions may arise, or workers may cease seeking matches on this platform. Is the proposed method applicable in such scenarios? Furthermore, do the regenerated workers and jobs retain their original attributes or latent factors?
2.In Chapter 3, "A Two-Stage Model for Multi-Category Matching," and in the description of the "MULTI-CENTROID GROUPING PENALTY" in Chapter 4, some complex steps and reasoning are intertwined. To facilitate clearer reading, it is recommended to streamline the pseudocode for presentation.
3.The matching system's attempts to improve the success rate of kidney exchange are impressive. Could the authors provide additional practical details and more experimental results—beyond just the accuracy of matching predictions—as well as insights into the system's effectiveness in other applications? This supplementary content could be included in an appendix.

---

### Official Review · Reviewer_rzsr · 2024-10-26

**Soundness:** 2
**Presentation:** 3
**Contribution:** 2
**Rating:** 5
**Confidence:** 4

**Summary:**

This paper addresses dynamic matching problems on platforms where supply (jobs) and demand (workers) fluctuate and attributes are often uncertain. The authors propose a two-stage latent factor model to improve matching efficiency: in the first stage, historical data is used to assign latent feature vectors to workers and jobs; in the second, these vectors are used for real-time dynamic matching. The model incorporates a multi-center grouping penalty to avoid overfitting and effectively cluster workers and jobs by type. Simulation results and a case study using kidney exchange data demonstrate that this approach provides improvements over baseline methods.

**Strengths:**

1. The paper provides theoretical guarantees for predictive consistency, group robustness, and convergence rates. These guarantees help establish confidence in the model's adaptability and performance.

2. The use of real kidney exchange data highlights the model’s adaptability to
certain real-world applications. By designing a model that considers medical and operational constraints, the authors showcase the potential of this approach in critical matching problems beyond traditional platforms.

3. The grouping penalty mechanism adds flexibility to the model. It prevents overfitting while ensuring that worker-job pairs are optimized for skill compatibility.

**Weaknesses:**

1. [Scalability to large scales]: In the simulations, only a limited number of worker types (|I∣=4) and job types (∣J∣=3) are tested (Section 6.1). The statistics of the kidney exchange dataset is missing in the paper. Real-world platforms (e.g., gig economy platforms, online job recommender systems) would require handling significantly more types of jobs and workers. Without evidence of scalability tests with larger worker-job type spaces, it’s uncertain if the model can handle the complexity of diverse real-world data effectively, particularly as computation time could increase exponentially with more latent factors.

2. [Lack of comprehensive comparison]: In Section 6.3, the authors describe the point process model as an alternative but do not provide direct performance comparisons such as AUC, accuracy, or computational efficiency on the same kidney dataset. Additionally, there is limited analysis on how each model performs with temporal data, given that point process models are often preferred for event-driven, time-sensitive scenarios.

3. [Data noise and imbalance]: The model assumes labeled data for training latent factors and matching, but in practical applications, data is often noisy or imbalanced, as seen in fields like crowdsourcing and organ donation w
here certain worker types or job types (e.g., rare blood types in kidney exchanges) are underrepresented. The lack of strategy for handling label noise or class imbalance may lead to reduced performance or biased matching results, especially for minority classes.

**Questions:**

1. How does the model handle situations where worker or job types exhibit highly irregular regeneration patterns or unpredictable availability?


2. How does the model work on scenarios where there are large n
umber of worker and job types in online web platforms (e.g., some online recommender systems)?

**Details Of Ethics Concerns:**

The kidney exchange dataset used includes sensitive information such as patient blood type, age, gender, and other demographic details. The authors should at least discuss or demonstrate how they deal with
such ethics concern in the paper.

---

### Official Review · Reviewer_1Fjb · 2024-10-28

**Soundness:** 2
**Presentation:** 2
**Contribution:** 2
**Rating:** 3
**Confidence:** 4

**Summary:**

This paper proposes a new method for dynamic supply-demand matching on platforms by utilizing a latent factor model combined with a multi-centroid grouping penalty mechanism. The authors introduce a two-stage model to handle the uncertainty and variability of worker attributes, enhancing the efficiency and adaptability of the matching process. In the first stage, latent feature vectors for workers and jobs are generated and grouped based on historical data. In the second stage, these learned features are used to dynamically match workers and jobs. Simulation experiments and a real-world case study using kidney exchange data demonstrate the model's effectiveness, showing that it can outperform traditional point process models in dynamic matching scenarios.

**Strengths:**

This paper presents a unique and well-structured dynamic matching approach that combines a latent factor model with a multi-centroid grouping penalty to handle uncertainty in user attributes, thereby improving matching efficiency and adaptability. The model is theoretically robust, with proofs of predictive consistency, and is validated through simulations and real-world kidney exchange data, demonstrating stronger performance than traditional methods.

**Weaknesses:**

The paper lacks a comprehensive review of related literature, resulting in weak research motivation. Without placing this work within the context of existing studies, it fails to highlight the specific challenges and significance of the problem. Additionally, only one real-world dataset (kidney exchange) is used in the experiments, which limits the demonstration of the model's generalizability and its relevance to the defined problem. The context gap between the problem statement and the dataset also raises questions about applicability. Furthermore, the paper does not compare with recent baseline models, which could have substantiated the effectiveness and robustness of the proposed modules and strategies.

**Questions:**

Section 6.1 mentions the FLGP model. Is this model the same as the point process model mentioned in the abstract? It is recommended to add an explanation of this model.
To improve the paper, the authors are encouraged to address the weaknesses outlined, including expanding the literature review, clarifying the model’s unique aspects, and providing comparisons with recent methods and additional experimental analysis.

---

### Official Review · Reviewer_qa9V · 2024-10-30

**Soundness:** 1
**Presentation:** 2
**Contribution:** 1
**Rating:** 3
**Confidence:** 3

**Summary:**

This paper proposes a latent factor model for the dynamic matching problem with an eye towards supply and demand balancing problems.  The proposed model incorporates a multi-centroid grouping penalty to handle heterogeneous workers and jobs.  The proposed approach operates in two phases: (1) grouping workers and jobs using historical data to establish latent feature vectors and (2) dynamically matching workers to jobs based on these features.

The paper provides theoretical guarantees in terms of predictive consistency, group robustness, and convergence rate.  In numerical studies, the paper considers a simulation of a matching system and a kidney transplant case study.

**Strengths:**

- The use of the multi-centroid grouping penalty is an appealing solution for the type of clustering problem (over worker and job types) that is being considered in this paper.
- The introduction is well-written, and motivates the study of the dynamic matching problem under a learning framework fairly well.
- I appreciate the effort to include theoretical results that describe many of the properties that one would be interested in when considering a proposed model.

**Weaknesses:**

- While the earlier parts of the paper are well-written, the later sections feel somewhat rushed and confusing at parts.  See below for my specific comments.
-  The notation used throughout the paper is somewhat difficult to understand, and there are inconsistencies.  For example, the sets associated with the group membership of workers and jobs are introduced as $\mathbf{V}$ and $\mathbf{U}$, but then subsequently referred to using $V$ and $U$, respectively.  Fixing these and any other inconsistencies (e.g., $A \leftrightarrow \mathbf{A}$, $B \leftrightarrow \mathbf{B}$, $L \leftrightarrow \mathcal{L}$, $G \leftrightarrow \mathcal{G}$) would help with the paper’s readability.  Streamlining notation wherever possible would also be appreciated.
- While I did not spend a large amount of time looking at them, the proofs in the Appendix seem to skip several steps.  E.g., on line 655, the proof of Theorems 1 and 3 claims “With the choice of $\lambda_n$ as mentioned, we can show: $\Vert \hat{\Theta} - \Theta^\star \Vert = O_p (n^{-1/2})$”, which implies the theorem statement.  However, $\Vert \hat{\Theta} - \Theta^\star \Vert$ was not mentioned anywhere before this statement in the proof, so it is a little unclear where this is coming from.  Similarly, on line 692, the proof claims that “And as $n$ approaches positive infinity, $R(\hat{V},  \hat{U})$ goes to $1$”.  The justification for this claim is not clear from the preceding lines.
- The experimental evaluation in Section 6.1 feels somewhat lacking to me — see specific comments below:
    - The last sentence of Section 6.1 states “We found that the LFGP model performs well in handling binary labels, but its accuracy decreases when dealing with more label types.”  I appreciate the acknowledgement of the model’s limitations, but I do not see the corresponding evidence for this claim?  In the setup, I do not see any characterizations of the label types considered that would suggest how this comparison and conclusion was reached.
    - The setting considered seems small and somewhat arbitrary when compared with a real application to matching markets.
- In Section 6.2, I appreciate the increased detail in the setup description for the real-world data.  Including evaluations on this data set somewhat addresses my concerns with the experiment in Section 6.1.
    - However, I am a little confused about the amount of detail dedicated to evaluating the LFGP model on this task — as far as I can tell, the only reported performance metric for LFGP on the kidney exchange task is in the sentence “We applied the processed dataset to the model and obtained an accuracy of 60% for matching prediction”.
    - The remainder of Section 6 describes an application of the point process model to the same problem as a comparison with LFGP.  The explanation of this implementation seems lengthy, and could be relegated to an Appendix.
    - In contrast, the comparison between the point process model and LFGP is isolated to a single paragraph, namely lines 506-516.  Since the performance of the point process model is reported as a ROC curve, and the performance of LFGP is reported as a single accuracy number (60%), this comparison is confusing and inconclusive.
- I would expect to see significantly more results in a quantitative sense (e.g., figures, performance metrics that are the same for the proposed model and the comparison model) to support the claims made in the abstract and introduction.
- It seems that [Xu et al., 2023 (10.1080/01621459.2023.2178925)] use the multi-centroid grouping penalty as part of a two-stage model for multi-categorical crowdsourcing in the context of a platform such as Amazon MTurk, where the first stage is to fit the observed labels with a latent factor model, and the second stage is to identify high-quality workers and to make predictions.  It is unclear to me how much of the current work is a new contribution with respect to this paper.

**Questions:**

- I had to read Section 2 a couple times to understand it — in paragraph 4, where the paper introduces $\mathbf{R}$ and $\mathbf{Z}$, it may be worth adding more detail to clarify what the labels mean in this application, and how to interpret them.
- The flow of the description in Section 3, Stage 2 (dynamic matching) could be improved — e.g., after the first paragraph, it would be nice to understand the different workers that must be considered and why they are different.  Introducing notation here may be useful to better contextualize the details that follow.
- The figures in Section 6 are hard to read.   The legends/labels should be significantly larger.
- If I understand the paper correctly, $C$ and $D$ (the number of job and worker clusters) are hyperparameters of your framework.  It would be interesting to understand what happens in your clustering approach as this hyperparameter varies -- i.e., how well does the matching perform if $C$ and $D$ do not reflect some ground truth?
- How is the dynamic matching affected if a new worker arrives, i.e., one with no historical record?  I believe it would be worth spending a few words in the text to discuss this case and how it would be handled by the proposed approach.
- How well does the latent factor model handle distribution shift?  E.g., if the jobs/workers in the second state (matching stage) don’t resemble the workers in the first stage?  The introduction suggests that there may be a “lifelong learning” component to your model that continually updates these latent factors, but I do not see any corresponding discussion in the evaluation.
- Please detail this paper's contributions with respect to [Xu et al., 2023 (10.1080/01621459.2023.2178925)].

---

### Official Review · Reviewer_QdpK · 2024-11-02

**Soundness:** 3
**Presentation:** 2
**Contribution:** 2
**Rating:** 3
**Confidence:** 3

**Summary:**

The paper investigates the supply-demand matching problem on dynamic platforms. The paper proposes a latent factor model and multi-centroid grouping penalty mechanism to perform dynamic matching. It includes two stages: the first stage predicts latent factors in historical data; the second stage utilize these features for dynamic matching. The simulation experiments and a real case study demonstrate effectiveness of the approach. However, my biggest concern about this paper is that contribution compared to the existing work and novelty of the approach is not clear.

**Strengths:**

+ The problem of supply-demand matching in dynamic settings is important and has practical implications, particularly in labor markets and organ transplantation.
+ The approach has theoretical guarantee.
+ The experiments conducted on both simulation and real case study further verify the effectiveness the approach.

**Weaknesses:**

- The contribution of the approach is not sufficiently clear. What is the difference between this paper and existing work? The paper even does not include related work section.
- The approach is straightforward and lacks novelty. The latent factor learning is straightforward and sort of trivial.
- The writing could be better. The paper should add adequate contents to clarify the difference between this work and the existing literature.

**Questions:**

Please refer to weaknesses.

---

### Note · Authors · 2024-11-26

I have read and agree with the venue's withdrawal policy on behalf of myself and my co-authors.